# Effect of Thrombin on the Metabolism and Function of Murine Macrophages

**DOI:** 10.3390/cells11101718

**Published:** 2022-05-23

**Authors:** Ürün Ukan, Fredy Delgado Lagos, Sebastian Kempf, Stefan Günther, Mauro Siragusa, Beate Fisslthaler, Ingrid Fleming

**Affiliations:** 1Institute for Vascular Signalling, Centre for Molecular Medicine, Goethe University, 60596 Frankfurt am Main, Germany; ukan@vrc.uni-frankfurt.de (Ü.U.); lagos@vrc.uni-frankfurt.de (F.D.L.); kempf@vrc.uni-frankfurt.de (S.K.); siragusa@vrc.uni-frankfurt.de (M.S.); fisslthaler@vrc.uni-frankfurt.de (B.F.); 2Bioinformatics and Deep Sequencing Platform, Max Planck Institute for Heart and Lung Research, 61231 Bad Nauheim, Germany; stefan.guenther@mpi-bn.mpg.de; 3German Center of Cardiovascular Research (DZHK), Partner Site RheinMain, 60596 Frankfurt am Main, Germany; 4CardioPulmonary Institute, Goethe University, 60596 Frankfurt am Main, Germany

**Keywords:** SMOC1, macrophage polarization, thrombin

## Abstract

Macrophages are plastic and heterogeneous immune cells that adapt pro- or anti-inflammatory phenotypes upon exposure to different stimuli. Even though there has been evidence supporting a crosstalk between coagulation and innate immunity, the way in which protein components of the hemostasis pathway influence macrophages remains unclear. We investigated the effect of thrombin on macrophage polarization. On the basis of gene expression and cytokine secretion, our results suggest that polarization with thrombin induces an anti-inflammatory, M2-like phenotype. In functional studies, thrombin polarization promoted oxLDL phagocytosis by macrophages, and conditioned medium from the same cells increased endothelial cell proliferation. There were, however, clear differences between the classical M2a polarization and the effects of thrombin on gene expression. Finally, the deletion and inactivation of secreted modular Ca^2+^-binding protein 1 (SMOC1) attenuated phagocytosis by thrombin-stimulated macrophages, a phenomenon revered by the addition of recombinant SMOC1. Manipulation of SMOC1 levels also had a pronounced impact on the expression of TGF-β-signaling-related genes. Taken together, our results show that thrombin induces an anti-inflammatory macrophage phenotype with similarities as well as differences to the classical alternatively activated M2 polarization states, highlighting the importance of tissue levels of SMOC1 in modifying thrombin-induced macrophage polarization.

## 1. Introduction

The coagulation cascade and the innate immune system are activated in response to insult or injury and act to stop blood loss; eradicate invading pathogens; and, later on, to re-establish homeostasis. Despite evidence of collaboration between the two pathways, humoral connections between blood constituents and inflammatory macrophages are not well understood. One potential molecular mediator of such communication is the serine protease thrombin [1]. Best known for its actions on fibrinogen and platelets, thrombin exerts its effects via the activation of G-protein-coupled receptors belonging to the protease activated receptor (PAR) family [2]. Four different PARs are expressed by different cell types, and thrombin interacts with PAR1, 3, and 4 to initiate downstream actions [3]. The best studied are probably the PAR1- and PAR4-mediated effects on platelet aggregation [4], but thrombin plays an important nonhemostatic role in the disruption of endothelial cell barrier function [5] and endothelial cell activation, leading to monocyte adhesion [6,7] and even cancer [8,9].

PARs are also expressed by monocytes where their activation has been linked with atherosclerosis [10,11], but the consequences of thrombin on macrophage responses are unclear. Indeed, in mice, thrombin has been reported to activate PAR2 to promote macrophage polarization into a classically activated or M1-like phenotype to induce inflammatory responses [12,13] and to sensitize macrophages to M1 polarization induced by interferon-γ (IFN-γ) [14]. However, thrombin has also been reported to induce alternative or M2 macrophage polarization with impaired plasticity [15,16]. There are a number of proteins known to inhibit thrombin activity, but recently, a matricellular protein, i.e., secreted modular Ca^2+^-binding protein (SMOC 1), was reported to bind to and activate thrombin to promote platelet activation [17]. The aim of the present study was, therefore, to revisit the topic of thrombin-induced macrophage polarization and to determine how SMOC1 affects thrombin-induced macrophage polarization.

## 2. Materials and Methods

### 2.1. Animals

Wild-type (C57BL/6) mice were from Charles River (Sulzfeld, Germany), SMOC1 (B6D2-Smoc1 < Tn(sb-lacZ,GFP)PV384Jtak > /JtakRbrc) mice (SMOC1^+/−^) were from the RIKEN BioResource Center (Tsukuba, Japan), and mTnG (B6.129(Cg)-Gt(ROSA)26Sortm4(ACTB-tdTomato,-EGFP)Luo/) mice were kindly provided by Ralf Adams (Münster). All animals were housed in conditions that conform to the Guide for the Care and Use of Laboratory Animals published by the U.S. National Institutes of Health (NIH publication no. 85-23). Both the University Animal Care Committee and the Federal Authority for Animal Research at the Regierungspräsidium Darmstadt (Hessen, Germany) approved the study protocol (V54-19c, 2.04.2020). For the isolation of bone marrow, mice were sacrificed using 4% isoflurane in air and cervical dislocation.

### 2.2. Monocyte Isolation and Culture

Murine monocytes: Monocytes were isolated from the bone marrow of 8–10-week-old mice and cultured in RPMI 1640 medium (Invitrogen, Karlsruhe, Germany, Baden-Wurtemberg) supplemented with M-CSF (15 ng/mL; Peprotech, Hamburg, Germany) and GM-CSF (15 ng/mL; Peprotech, Hamburg, Germany) for 7 days to generate macrophages.

### 2.3. Macrophage Polarization

Macrophages were polarized to classical activated M1 macrophages by treating with LPS (human: 100 ng/mL, murine: 10 ng/mL; Sigma-Aldrich, Munich, Germany) and IFN-γ (human: 20 ng/mL, murine: 1 ng/mL; Peprotech, Hamburg, Germany) for 12 h and to alternative M2a macrophages by treating with IL-4 (20 ng/mL; Peprotech, Hamburg, Germany) for 24 h. Pro-resolving M2c macrophages were repolarized from M1 macrophages by the addition of TGF-β1 (10 ng/mL; Peprotech, Hamburg) for 48 h. Thrombin-induced macrophage polarization was achieved by incubating macrophages with thrombin (0.1 or 1 U/mL; Haemochrom Diagnostica Essen, Germany) for up to 48 h.

### 2.4. Immunoblotting

Bone-marrow-derived macrophages were lysed in ice-cold Triton X-100 lysis buffer (20 mmol/L Tris/HCl (pH 7.5), 1% Triton X-100, 25 mmol/L β-glycerolphosphate, 150 mmol/L NaCl, 10 mmol/L Na pyrophosphate, 20 mmol/L NaF) containing 2 mmol/L Na orthovanadate, 10 mmol/L okadaic acid, and a protease inhibitor mix (2 μg/mL antipain, 2 μg/mL aprotinin, 2 μg/mL chymostatin, 2 μg/mL leupeptin, 2 μg/mL pepstatin, 2 μg trypsin inhibitor, and 40 μg/mL phenylmethysulfonylfluoride). Samples were then separated by SDS-PAGE and subjected to Western blotting. Detection was performed by enhanced chemiluminescence using a commercially available kit (Amersham, Freiburg, Germany) as described [18].

### 2.5. RT-qPCR

Total RNA was extracted and purified from wild-type and SMOC1+/− macrophages using Tri Reagent (ThermoFisher Scientific, Karlsruhe, Germany) according to the manufacturer’s instructions. RNA was eluted in nuclease-free water, and RNA concentration was spectrophotometrically determined at 260 nm using a NanoDrop ND-1000 (ThermoFischer Scientific, Karlsruhe, Germany). For the generation of complementary DNA (cDNA), total RNA (250 ng) was reverse-transcribed using SuperScript IV (ThermoFischer Scientific, Karlsruhe, Germany) and random hexamer primers (Promega, Madison, WI, USA) according to the manufacturer’s protocol. qPCR was performed using SYBR green master mix (Biozym, Hessisch Oldendorf, Germany) and appropriate primers (Table 1) in a MIC-RUN quantitative PCR system (Bio Molecular Systems, Upper Coomera, Australia). Relative RNA levels were determined using a serial dilution of a positive control. The data are shown relative to the mean of the housekeeping genes, elongation factor (EF) 2 and 18S RNA.

### 2.6. Phagocytosis Assays

Murine bone-marrow-derived monocytes were seeded onto 96-well plates and differentiated into macrophages by treating with MCSF and GMCSF for 7 days. After differentiation was complete, macrophages were polarized as described above, or treated with thrombin for up to 48 h. Thereafter, either dil-oxLDL (1:200; Thermofisher, Darmstadt, Germany) or pHrodo Green zymosan bioparticles (1 mg/mL, Thermofisher, Darmstadt, Germany) was added, and phagocytosis was monitored over 24 h using an automated live cell imaging system (IncuCyte; Sartorius, Göttingen Germany).

### 2.7. FACS Analysis

Detection of the pro-inflammatory cytokine, and chemokine secretion was performed according to the instructions of manufacturer (BioLegend, LEGENDplex Mouse Proinflammatory Chemokine Panel). In brief, macrophage supernatant was collected in 1.5 mL tubes, snap froze with liquid nitrogen, and kept in −80 °C until measurement. Samples and standards provided by the kit were transferred onto a 96-well filter plate, then incubated with equal amounts of assay buffer, and in beads for 2 h at room temperature on a shaker. Wells were washed with wash buffer twice, followed by an incubation with the detection antibody for an hour again on a shaker. At the end of incubation, SA-PE was added into the wells, and the plate was left to incubate for 30 min on a shaker; then, wells were again washed with wash buffer twice, and samples were re-suspended with wash buffer for immediate FACS analysis.

### 2.8. RNA Sequencing

Total RNA was isolated from macrophages by using an RNeasy Micro kit (Qiagen, Hilden, Germany) on the basis of the manufacturer’s instructions. The RNA concentrations were determined by using NanoDrop ND-1000 (TFS, Waltham, MA, USA; λ 600 nm). Total RNA (1 µg) was used as input for the SMARTer Stranded Total RNA Sample Prep Kit—HI Mammalian (Takara Bio, Kusatsu shi, Japan).

Trimmomatic version 0.39 was employed to trim reads after a quality drop below a mean of Q20 in a window of 20 nucleotides and keeping only filtered reads longer than 15 nucleotides [19]. Reads were aligned versus Ensembl mouse genome version mm10 (Ensembl release 101) with STAR 2.7.10a [20]. Aligned reads were filtered to remove duplicates with Picard 2.25.5 (Picard: A set of tools (in Java) for working with next-generation sequencing data in the BAM format), multi-mapping, ribosomal, or mitochondrial reads. Gene counts were established with featureCounts 2.0.2 by aggregating reads overlapping exons on the correct strand, excluding those overlapping multiple genes [21]. The raw count matrix was normalized with DESeq2 version 1.30.1 [22]. Contrasts were created with DESeq2 on the basis of the raw count matrix. Genes were classified as significantly differentially expressed at average count > 5, multiple testing adjusted *p*-value < 0.05, and −0.585 < log2FC > 0.585. The Ensemble annotation was enriched with UniProt data [23].

### 2.9. Metabolomics

Metabolites for targeted metabolomics were extracted by scraping the cells on ice, with ice-cold extraction buffer, containing methanol, formic acid (0.1%), TCEP (1 mmol/L), and sodium ascorbate (1 mmol/L). After centrifugation and removal of protein, the sample extract was divided into equal fractions for amino acid and TCA cycle analysis.

#### 2.9.1. Amino Acid Analyses

Samples were spiked with an internal standard mix and dried (Concentrator plus; Eppendorf, Hamburg, Germany) at 45 °C. The dried samples were reconstituted in 20 µL HCl (20 mmol/L) and derivatized according to the AccQ-Tag derivatization kit protocol (Waters GmbH, Eschborn, Germany) by adding 70 µL borate buffer and 20 µL ACQ-Tag reagent and incubating at 55 °C for 10 min. For data acquisition, an Agilent 1290 Infinity II ultra-performance liquid chromatography (UPLC) system coupled to a QTrap 5500 LC–MS/MS system from ABSciex (Darmstadt, Germany) was used in positive ion mode. Metabolite separation of derivatized amino acids was achieved with a flow rate of 300 µL/min at 35 °C on an Extend C18-column (150 × 2.1 mm, 1.8 µm; Agilent). Raw data extraction and peak identification was performed using the SCIEX OS (2.2) software.

#### 2.9.2. TCA Cycle

An internal standard mix was added to the corresponding fraction, and samples were freeze-dried (Alpha 3–4 LSCbasic; Christ, Osterode am Harz, Germany). Dried samples were reconstituted in formic acid (0.5%) and introduced to an Agilent 1290 Infinity UPLC platform coupled to an Agilent 6495 Triple quadrupole LC/MS system (Santa Clara, CA, USA) used in negative ion mode. Metabolites were separated on an ACQUITY UPLC HSS T3 column (150 × 2.1 mm, 1.8 µm; Waters) at a flow rate of 300 µL/min at 40 °C. Raw data extraction and peak identification were performed using the MassHunter Quantitative Analysis software.

Data analysis was performed after normalization to the protein content. For the unsupervised principal component analysis, MetboAnalyst 5.0 (www.metaboanalyst.ca (accessed on 22 March 2022)) was used.

### 2.10. Immunofluorescence

After polarizations were completed, macrophages were treated with diloxLDL 1:200 (Thermo Fischer Scientific) for 15 min, then fixed with 4% Rotifix, and blocked with 3% horse serum for an hour. At the end of incubation macrophages were incubated with primary antibodies overnight at +4 °C on a shaker, then washed 3 times with 1X PBS. After the washing was completed, macrophages were incubated with secondary antibodies for an hour on a shaker and washed again 3 times with phosphate-buffered saline. Finally, macrophages were incubated with Hoechst (1:1000), and images were taken with a confocal microscope (LSM-780; Zeiss, Jena, Germany) with ZEN Software (Zeiss).

### 2.11. Endothelial Cell Proliferation

Murine pulmonary endothelial cells were isolated from mTnG mice as described previously [18] and used at passage 5 following repurification with CD31-antibody-coated magnetic beads. Thereafter, cells (10^4^ cells/96-well) were seeded in DMEM/F12 containing glucose/L (5 mmol/L), 20% FCS, endothelial cell growth supplement with heparin, and penicillin and streptomycin (each 50 ug/mL). After complete adherence (4 to 6 h), the medium was exchanged to 50% MLEC basal medium (2% FCS but without ECGS-H) and 50% macrophage-conditioned medium. Pictures for phase contrast and GFP-emitted fluorescence were taken every 4 h with an InCucyteS3 (Sartorius, Göttingen, Germany) to monitor cell division and plate coverage. Green nuclei were normalized to the number of nuclei at 4 h.

### 2.12. Statistical Analyses

Data are expressed as mean ± SEM. One-way ANOVA was used for comparison of three or more groups with one variable, and two-way ANOVA was used for comparison of variance between multiple groups and two variables. ANOVA was followed by Bonferroni’s or Tukey’s multiple comparison tests. Values of *p* < 0.05 were considered statistically significant.

## 3. Results

### 3.1. Effects of Thrombin on Macrophage Gene Expression

Murine bone-marrow-derived macrophages were maintained in culture or treated with LPS and IFN-γ to generate classically activated M1 macrophages, with IL-4 to generate alternatively activated M2a macrophages, or with LPS and IFN-γ followed by TGF-β to result in pro-resolving M2c macrophages. The expression of routinely used polarization markers was then assessed and compared with the effects of incubating macrophages with thrombin (1 U/mL) for up to 48 h. Different time points of thrombin stimulation were assessed as the M1 response usually peaks at or before 12 h, while the pro-resolving polarization takes up to 48 h.

Thrombin–treated macrophages did not express classical M1 markers such as tumor necrosis factor-α (TNF-α), inducible nitric oxide synthase (iNOS), or the M2 marker gene found in inflammatory zone-1 (Fizz-1) (Figure 1A). Thrombin did, however, increase the expression of MRC-1, MMP9, and CD36, with slightly higher expression detected after 48 rather than 24 h. The lack of effect on iNOS was confirmed at the protein level, as was the inability to induce the phosphorylation of STAT6 (Figure 1B), which tends to be elevated in M2a-polarized macrophages [24]. Different cytokines are secreted by macrophages in different polarization states [25]. For example, CCL5 (RANTES) and CCL2 (monocyte chemoattractant protein-1) were produced by M1- and M2c-polarized macrophages, but high levels of CCL22 (macrophage-derived chemokine) were only generated by M2c macrophages. This contrasted with previous reports of it being elevated in M2a-polarized cells [26]. Thrombin did not elicit the expression of CCL2 or CCL5 but did increase CCL22 levels (Figure 1C). These observations indicate that thrombin elicits changes in macrophage gene expression that are consistent with a state intermediate between M2a and M2c.

### 3.2. Metabolic Characterization of Thrombin-Polarized Macrophages

Monitoring metabolism provides a more accurate indication of functional similarities than the analysis of specific marker genes. Therefore, TCA cycle pathway metabolites as well as amino acids were quantified in M0-, M1-, M2a-, M2c-, and thrombin-polarized murine macrophages. Principal component analysis revealed metabolic signatures that were clearly distinct in M0-, M1-, M2a-, and M2c-polarized cells (Figure 2A). Thrombin-treated macrophages were most similar to M2a-polarized cells after 24 h and most similar to the M0 state after 48 h. Metabolites of the TCA cycle were largely comparable in M2a- and thrombin-polarized macrophages (Figure 2B). An analysis of amino acid levels, however, revealed clear differences between the M2a- and thrombin-polarized groups (Figure 3A,B). While levels of citruilline, allantoin, and tryptophan were comparable in M2a- and thrombin-polarized macrophages (Figure 3C), levels of arginine, lysine, alanine, anthranilic acid, ethanolamine, and itaconate were clearly different. These metabolic data confirmed the results on the expression studies, i.e., that thrombin does not induce an M1-like activation of macrophages but rather reprograms them to a state with some similarities to M2 states.

### 3.3. Functional Characterization of Thrombin-Polarized Macrophages

Alternatively activated or M2 macrophages are regarded as a continuum of functionally and phenotypically related cells, with a critical role in type II inflammation and in the resolution and tissue repair phases [27]. These cells can be subdivided into M2a and M2b, corresponding to type-II-activated macrophages obtained by triggering Fcγ receptors in the presence of a Toll receptor stimulus, and M2c, which includes deactivation programs elicited by agents such as transforming growth factor β (TGF-β) [27]. There is even an M2d phenotype that results from adenosine-dependent “switching” of M1 [28].

One characteristic of M2 macrophages is their ability to clear apoptotic cell debris, as well as to take up different lipids by phagocytosis [29]. We, therefore, compared the phagocytosis of oxidized low-density lipoprotein (ox-LDL) by polarized murine macrophages and observed the expected low uptake by M1-polarized cells (Figure 4A). There was, however, a clear increase in phagocytosis following M2a and M2c polarization that was even surpassed by the effects seen in cells pretreated with thrombin for 48 h. A similar phenomenon was observed using pHrodo-labelled zymosan (Figure 4B,C). M2-polarized macrophages also release exosomes and a number of soluble factors to promote angiogenesis. Therefore, conditioned medium from polarized macrophages was collected and added to subconfluent pulmonary endothelial cells from mice expressing GFP in their nuclei. Conditioned medium from thrombin-treated macrophages clearly increased endothelial cell proliferation (Figure 4D) and was equally as effective as medium from M1- and M2a-polarized macrophages, which is consistent with the well-described effects of cytokines on angiogenesis [30].

### 3.4. Transcriptional Characterization of Thrombin-Polarized Macrophages

In order to gain insight into the impact of thrombin stimulation on alterations in macrophage gene expression, we performed RNA sequencing. While our metabolic data showed similar clustering between naïve, anti-inflammatory, and thrombin-polarized macrophages, the gene profile of thrombin-stimulated macrophages was clearly distinct from that of M0 macrophages (Figure 5A). For example, thrombin-polarized macrophages expressed higher levels of genes related to the TGF-β pathway, e.g., Id1, Id3, Smad6, and Smad9 than M0 cells. The expression of more inflammatory genes, e.g., Ifi213, Ifi44, Ifi206, Irf7, Tnfsf8, and Ccl5 was, on the other hand, lower, following stimulation with thrombin. There were also distinct differences between the expression profiles of M2a and thrombin-polarized cells, e.g., the classical M2a marker genes Il4i1, Arg1, Jak2, and Klf4 were significantly lower following thrombin stimulation (Figure 5B). GO term analysis revealed that while M2a macrophages favored pathways related to the cell cycle, thrombin-polarized macrophages expressed genes linked to responses to viruses and interferons as well as phagocytosis and engulfment (Appendix A).

### 3.5. Impact of SMOC1 on Thrombin Polarized Macrophages

SMOC1 was recently identified as a platelet-derived thrombin activator, and antibodies directed against the protein attenuated the thrombin-induced aggregation of murine and human platelets [17]. Given the close association between platelets and macrophages, we determined the impact of SMOC1 on macrophage gene expression and function. When gene expression was compared in macrophages treated with thrombin in the absence and presence of SMOC1 antibodies, we observed the differential regulation of several of the genes reported above. In particular, the expression of transcripts involved in TGF-β signaling (*Id1*, *Id3*, *Smad6*, *Smad9*) were all downregulated by the antibody, while members of the interferon signaling pathway increased (Figure 6A).

To determine the effect of SMOC1 on macrophage function, we focused on the phagocytosis of zymosan and ox-LDL. As was the case previously, the phagocytosis was highest in M2a- and M2c-polarized macrophages, but thrombin also stimulated phagocytosis (Figure 6B,C). Importantly, the addition of recombinant SMOC1 to macrophages amplified responses to thrombin, particularly to a low concentration of thrombin (0.1 U/mL), while an antibody directed against SMOC1 attenuated responses (Figure 6D). SMOC1−/− mice die shortly after birth [31,32], but the phagocytosis of zymosan induced by thrombin was attenuated in macrophages from SMOC1+/− mice and was rescued by the addition of recombinant SMOC1 (Figure 6E). Thus, the presence of SMOC1 has pronounced effects on the polarization of macrophages by thrombin.

## 4. Discussion

The results or our study indicate that thrombin elicits effects on macrophages that are distinct from the phenotype induced by classical activation protocols. Rather, thrombin stimulation resulted in macrophages adopting a pro-resolving phenotype characterized by the secretion of CCL22, a cytokine most akin to M2 and tumor-associated macrophages that impacted expression of genes of the TGF-β pathway. The manipulation of SMOC1 levels had a pronounced impact on the expression of TGF-β signaling-related genes as well as on macrophage function, highlighting the fact that tissue levels of SMOC1 can clearly modify thrombin-induced macrophage polarization.

The phenotypic characterization of macrophages is highly complicated, and there are many more distinct genetic fingerprints and metabolic states than are reflected in a basic M0/M1/M2 classification [33,34]. There is even still controversy about the cellular markers for each of the reported phenotypes and how in vitro polarization studies can be compared with the in vivo situation [35]. Such discrepancies in markers or the timing of the experiments may account for the previous reports that thrombin stimulation results in the M1-like [12,13,14] as well as M2-like polarization of macrophages [15]. In the current study, we characterized macrophages by virtue of classical polarization marker levels as well as by their ability to phagocytose extracellular material and to promote endothelial cell proliferation. The expression of genes regularly used to identify polarization subtypes indicated that thrombin induced an M2a-like polarization. For example, we observed comparable levels of CD36, which is a scavenger receptor for oxLDL uptake [36], and MMP9, which is also highly expressed by M2-type macrophages [37], in M2a- and thrombin-treated macrophages. However, levels of Fizz-1 were clearly induced by M2a polarization but unaffected by thrombin.

Changes in macrophage polarization are linked with changes in cell metabolism, not only that these alterations affect energy metabolism and biosynthesis but also influence immune function of the resulting macrophage phenotype. For example, pro-inflammatory macrophages display enhanced glycolysis and impaired TCA cycle function to meet with their metabolic needs and regulate the production of reactive oxygen species and inflammasome formation (for a review, see reference [38]). Conversely, alternatively activated M2 macrophages rely on oxidative phosphorylation and the TCA cycle to promote tissue remodeling and the reestablishment of homeostasis. Changes in arginine and citrulline levels generally reflect alterations in the activity of iNOS and arginase. Thus, the decreased citrulline and increased arginine levels in thrombin-polarized macrophages contrasts clearly with the M1 phenotype and has more similarity with M2a-polarized cells. However, there were some clear differences between M2a and thrombin-treated macrophages, and our finding that ethanolamine was higher in thrombin-polarized than in classical M2a-polarized macrophages may reflect their increased ability to phagocytose ox-LDL. Indeed, ethanolamine is the most frequent head group present in mammalian plasmalogens [39], and membrane ethanolamine plasmalogen deficiency has been reported to result in a decreased phagocytosis capacity [40]. There were also differences in itaconate, which is generated by diverting aconitate away from the TCA cycle during inflammatory macrophage activation and has been reported to link cell metabolism with stress and immune responses [41]. We found that itaconate levels were higher in more transitional phenotypes, i.e., repolarized M2c macrophages as well as thrombin-stimulated macrophages. Functionally, we focused on the ability of macrophages to phagocytose ox-LDL and zymosan, as well as the effects of the macrophage supernatant on endothelial cell growth, two classical functions of M2-polarized macrophages. In all of these assays, the function of M2a and thrombin-treated macrophages were similar, but there was a tendency for thrombin-treated macrophages to take up ox-LDL more efficiently.

RNA sequencing, however, clearly revealed distinctions between the M0 and M2a phenotypes and that induced by thrombin, including altered TGF-β signaling. Indeed, the expression of *Id1* and *Id3*, which partner with TGF-β to regulate cell proliferation and survival [42], were increased in thrombin-stimulated versus naïve M0 macrophages, as were *Smad6* and *Smad9*, which are inhibitors of the TGF-β [43] and BMP [44] pathways, respectively. In agreement with the phenotypic and metabolic data, thrombin decreased the expression of bona fide pro-inflammatory genes, several of them related to IFN-γ (*Ifi213*, *Ifi44*, *Ifi206*, *Irf7*), TNF-α (*Tnfsf8*), and chemokine (*Ccl5*) pathways. Moreover, several genes that drive polarization towards M2a such as *Arg* [45], *Il4i1* [46], *Klf4* [47], *Jak2* [48], and *Socs2* [49] were clearly attenuated by thrombin stimulation.

In addition to our focus on clarifying the effects of thrombin on macrophage polarization, we set out to determine the impact of SMOC1 on thrombin-induced responses. SMOC1 is a matricellular protein that is generally localized to the basement membrane of different tissues [50,51,52,53]. Little is known about its role in physiology or pathophysiology, but it has been associated with both Waardenburg anophthalmia syndrome [31,32] and Alzheimer’s disease [54,55,56]. More recently, SMOC1 was identified as a glucose-responsive hepatokine important for glycemic control in mice [57], a phenomenon that could not be confirmed in a human population [58]. Rather, circulating levels of SMOC1 were found to increase in subjects with type 2 diabetes in parallel with platelet hyperactivity to thrombin [17]. To date, nothing is known about effects of SMOC1 in macrophages, but SMOC1 expression is at least partly regulated by miR-223 [59], and decreased levels of miR-223 in monocytes/macrophages have been linked with atherosclerosis and macrophage activation [60,61,62], as well as the transition between inflammation and cancer [63,64]. Given that SMOC1 expression is regulated by miR-223 [59], which when secreted can potentiate the actions of thrombin [17], it was tempting to suggest that the combination of thrombin and SMOC1 could affect innate immune responses. We found that SMOC1 did, indeed, increase phagocytosis by thrombin-treated macrophages, and that effects were attenuated in SMOC1+/− mice as well as by antibodies directed against SMOC1 and rescued by the addition of recombinant SMOC1 protein. All of these observations indicate that the macrophage expression of SMOC1 has a major impact on thrombin-induced macrophage polarization. At the level of gene expression, antibodies directed against SMOC1 reversed the thrombin-induced changes in a subset of genes, particularly those relating to TGF-β signaling and inflammatory interferon signaling. The overall effect was that the inactivation of SMOC1 in macrophages curbed the anti-inflammatory effects of thrombin.

Taken together, we have shown that thrombin stimulation polarizes macrophages to a distinct phenotype that initially seemed to be closest to the alternatively activated M2a phenotype. At first sight, this observation was unexpected, given the pro-inflammatory role attributed to thrombin in atherosclerosis [65] and reports that negative regulators of thrombin have athero-protective effects. For example, heparin cofactor II can protect elderly persons against carotid atherosclerotic lesions and hirudin; another thrombin inhibitor has been shown to decrease restenosis after angiography (for a review, see reference [66]). However, an M2a-like polarization of macrophages has previously been linked to a pro-fibrotic state, which could impact on wound healing [16]. Moreover, there is a well-established link between thrombosis and cancer [8], where thrombin-induced PAR1 signaling has been suggested to promote the immunosuppressive microenvironment that protects the tumor against host antitumor immune responsiveness [67]. There are also links between thrombin and epithelial–mesenchymal transition in several cancer cells, which is a key process implicated in cancer invasion and metastasis [68,69]. SMOC1 levels are elevated in some forms of cancer [70,71,72], and given that the cleavage of osteopontin by thrombin initiates its tumor promoting activity [73], it will be interesting to determine whether or not the interaction between SMOC1 and thrombin underlies some of this tumor-promoting ability.

## Figures and Tables

**Figure 1 cells-11-01718-f001:**
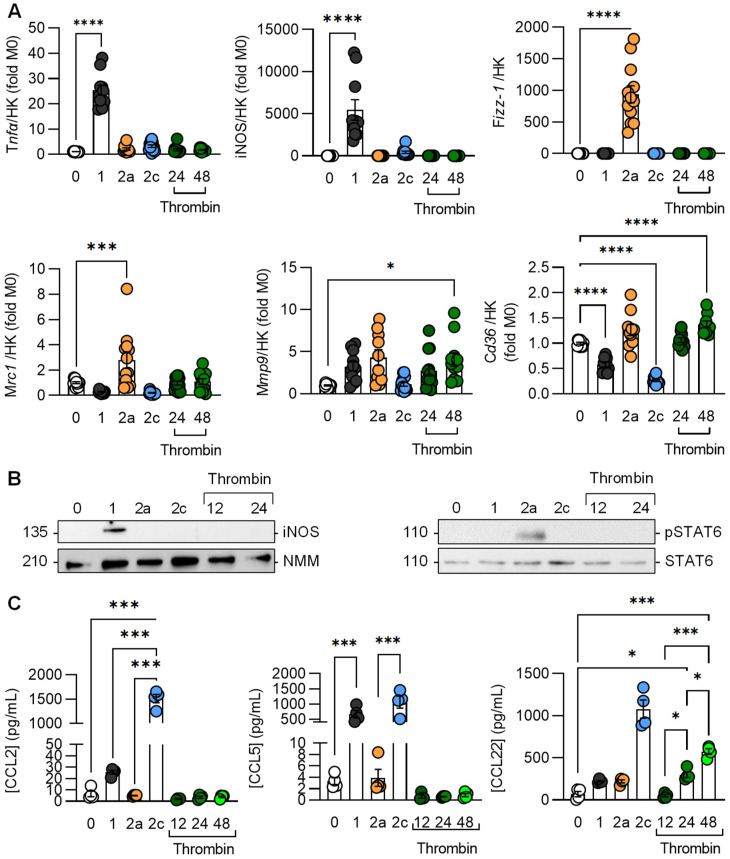
Effects of thrombin in murine macrophage chemokines. Murine bone-marrow-monocyte-derived macrophages were treated with either solvent (MØ) or with the combination of LPS (10 ng/mL) and INF-γ (1 ng/mL) for M1 polarization; IL-4 (20 ng/mL) for M2a polarization; and thrombin (1 U/mL) for 12, 24, and 48 h for Mth polarization. M1 macrophages were treated with TGF-β1 (10 ng/mL) for 48 h for M2c polarization. (**A**) mRNA expression of macrophage phenotype markers with qPCR; n = 6 mice. HK = housekeeping genes. (**B**) Representative blots showing the expression of iNOS and the phosphorylation of STAT6 (pSTAT6) in differently polarized macrophages; n = 4 mice. NMM = non muscle myosin. (**C**) Cytokine levels in the macrophage supernatant; n = 4 mice. All experiments were performed at least twice. (**A**,**C**) One-way ANOVA followed by Tukey’s multiple comparison test. * *p* < 0.05, *** *p* < 0.001, **** *p* < 0.0001.

**Figure 2 cells-11-01718-f002:**
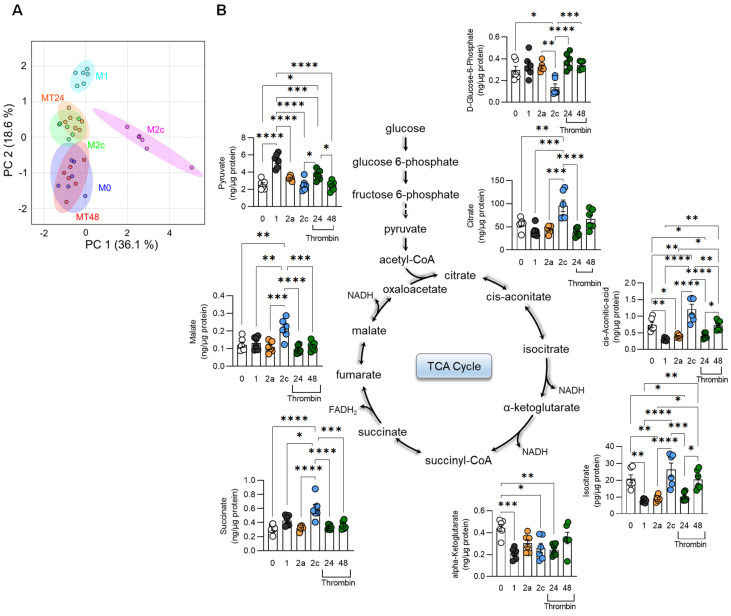
Effects of thrombin on macrophage TCA cycle metabolites. (**A**) Principal component (PC) analysis of TCA and glycolysis-related metabolites of the different macrophage polarization states versus cells treated with thrombin for 24 (MT24) or 48 h (MT48). (**B**) Polarization-dependent changes in specific glycolytic pathway and TCA cycle metabolites; n = 6 mice (one-way ANOVA and Tukey’s multiple comparison test). * *p* < 0.05, ** *p* < 0.01, *** *p* < 0.001, **** *p* < 0.0001.

**Figure 3 cells-11-01718-f003:**
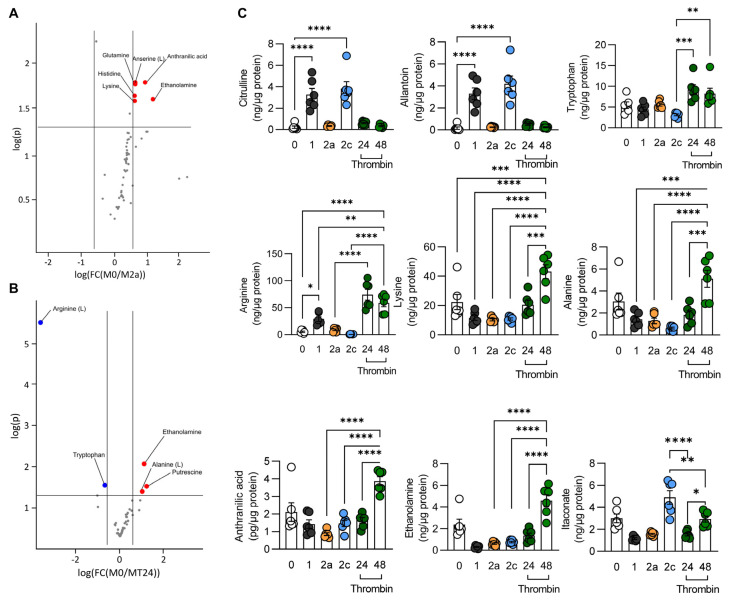
Effects of thrombin on macrophage amino acid metabolism. (**A**) Volcano plot comparing amino acid levels in M0 versus M2a macrophages. (**B**) Volcano plot comparing amino acid levels in M0 macrophages compared to macrophages treated with thrombin for 24 h (MT24). (**C**) Levels of selected amino acids in M0-, M1-, M2a-, and M2c-polarized macrophages versus macrophages treated with thrombin for up to 48 h; n = 6 mice (one-way ANOVA and Tukey’s multiple comparison test). * *p* < 0.05, ** *p* < 0.01, *** *p* < 0.001, **** *p* < 0.0001.

**Figure 4 cells-11-01718-f004:**
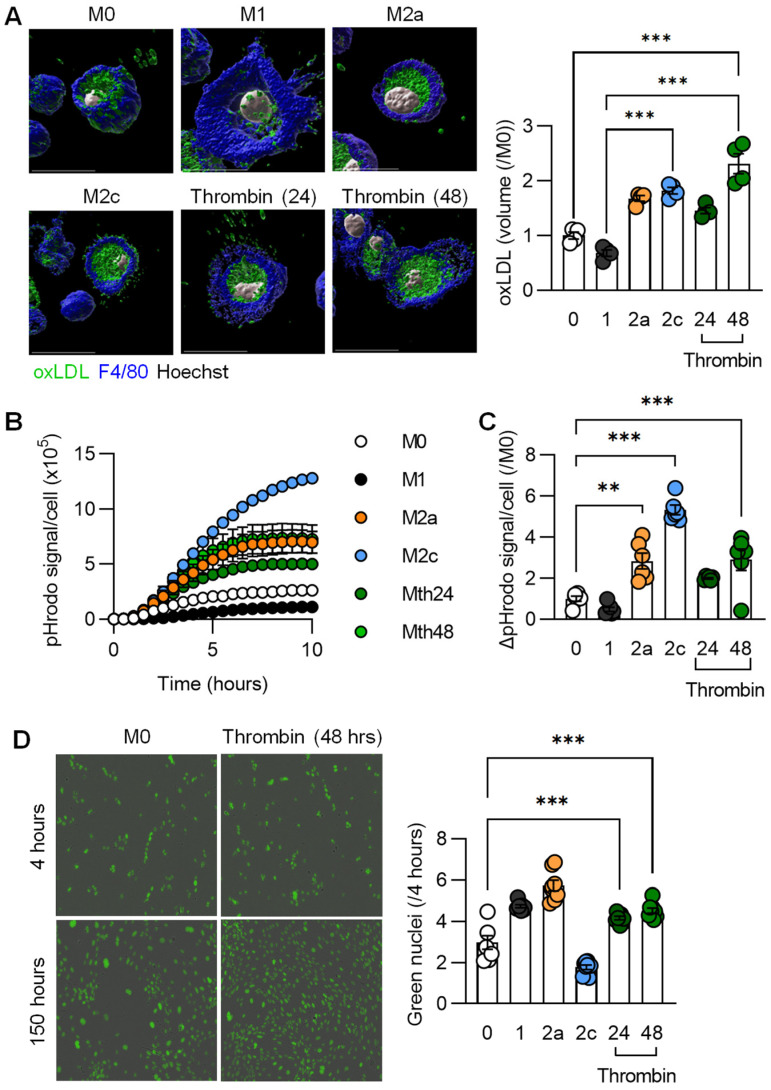
Consequences of thrombin polarization on macrophage function. Murine bone-marrow-monocyte-derived macrophages (M0) were either treated with solvent; polarized to the M1, M2a, or M2c phenotypes; or incubated with thrombin for up to 48 h. (**A**) Volume of dil-labeled oxLDL taken up over 15 min; n = 4 mice per group. (**B**) Time course of the phagocytosis of pHrodo-labaled particles over 10 h; n = 6 mice. (**C**) Quantification of phagocytic cells per field after 5 h; n = 6 mice. (**D**) Endothelial cell proliferation assay upon treatment with thrombin-polarized macrophage supernatants; n = 4 independent cell batches/mice. All experiments were performed at least twice. (**A**,**C**,**D**) One-way ANOVA and Bonferroni’s multiple comparisons test. ** *p* < 0.01, *** *p* < 0.001.

**Figure 5 cells-11-01718-f005:**
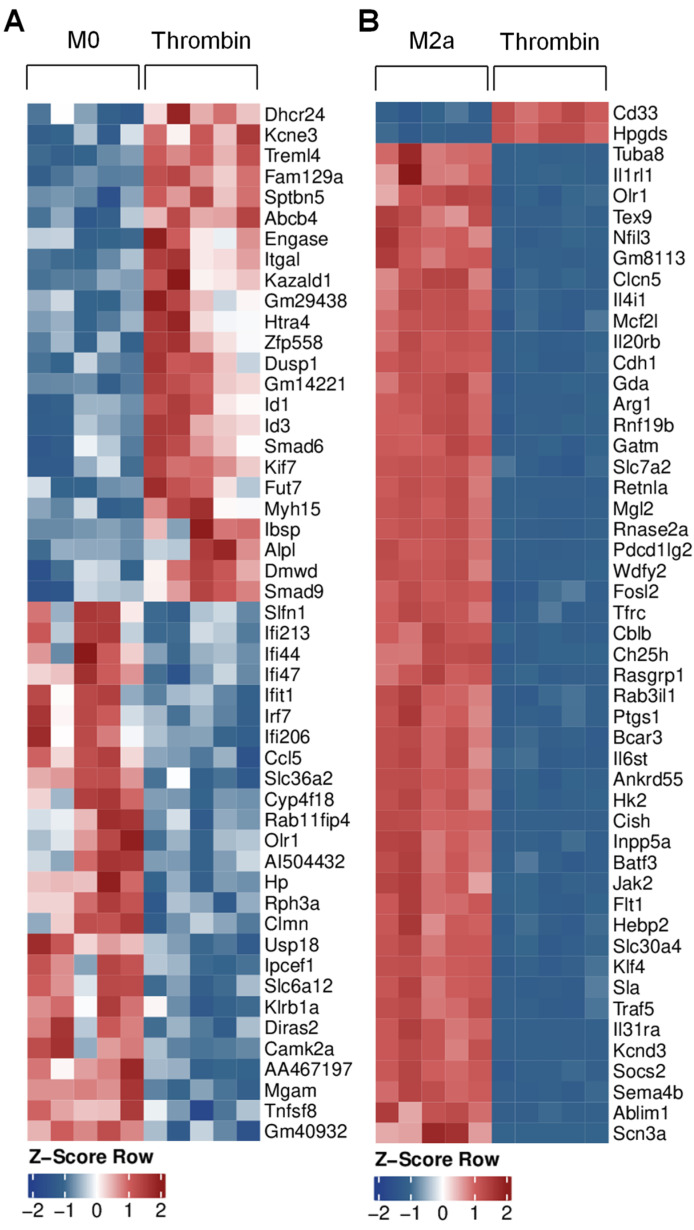
Thrombin-induced changes in macrophage gene expression. Murine bone-marrow-monocyte-derived macrophages (M0) were either treated with solvent, polarized to the M2a phenotype, or incubated with thrombin (0.1 U/mL, 24 h). (**A**) Top 50 differentially regulated genes between M0 and thrombin-treated macrophages; n = 5 mice. (**B**) Top 50 differentially regulated genes between M2a and thrombin-treated macrophages; n = 5 mice per group.

**Figure 6 cells-11-01718-f006:**
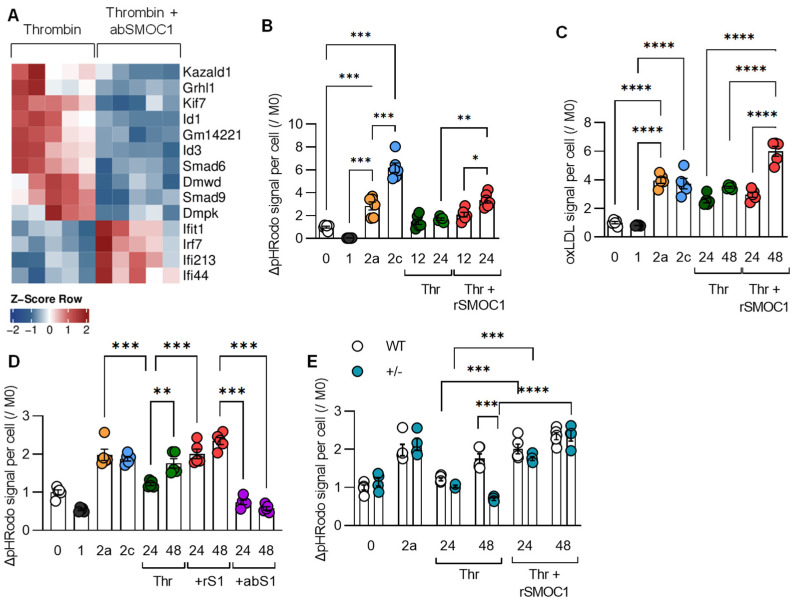
Modulation of thrombin-induced macrophage polarization by SMOC1. (**A**) Differentially regulated genes in murine bone-marrow-monocyte-derived macrophages treated with thrombin (0.1 U/mL, 24 h) in the absence and presence of an antibody directed against SMOC1 (1 µg/µL); n = 5 mice. (**B**) Phagocytosis of pHrodo zymosan in M0-, M1-, M2a-, and M2c-polarized macrophages compared to thrombin stimulation (Thr; 1 U/mL, up to 24 h) in the absence and presence of recombinant SMOC1 (rSMOC1; 0.5 µg/mL). n = 3 mice with all experiments preformed in duplicate. (**C**) Uptake of dil-oxLDL under the same conditions as in (B); n = 5 mice. (**D**) Phagocytosis of pHrodo zymosan in M0-, M1-, M2a-, and M2c-polarized macrophages compared to stimulation by thrombin (0.1 U/mL, up to 24 h) in the absence and presence of recombinant SMOC1 (rS1; 0.5 µg/mL) or an antibody directed against SMOC1 (abS1; 1 µg/µL); n = 5 mice. (**E**) Phagocytosis of pHrodo zymosan in M0- and M2a-polarized macrophages from wild-type (WT) and SMOC1+/− mice (+/−) compared to stimulation by thrombin (0.1 U/mL, up to 24 h) in the absence and presence of recombinant SMOC1 (rSMOC1; 0.5 µg/mL); n = 5 mice. Experiments (B-D) were performed at least twice. One-way ANOVA and Bonferroni’s multiple comparisons test (B–D), and two-way ANOVA and Bonferroni’s multiple comparisons test (E). * *p* < 0.05, ** *p* < 0.01, *** *p* < 0.001. **** *p* < 0.0001.

**Table 1 cells-11-01718-t001:** PCR primers used.

Gene	Forward	Reverse
NOS	GTGGTGACAAGCACATTTGG	GTTCGTCCCCTTCTCCTGTT
TNF-α	GGCCTTCCTACCTTCAGACC	CCGGCCTTCCAAATAAATAC
FIZZ-1	CCCTTCTCATCTGCATCTCC	CAGTAGCAGTCATCCCAGCA
CD206	TGGATGGATGGGAGCAAAGT	GCTGCTGTTATGTCTCTGGC
MMP9	GAAGGCAAACCCTGTGTGTT	AGAGTACTGCTTGCCCAGGA
CD36	AAACCCAGATGACGTGGCAA	AAGATGGCTCCATTGGGCTG
EF2	GACATCACCAAGGGTGTGCAG	GCGGTCAGCACACTGGCATA
18S	CTTTGGTCGCTCGCTCCTC	CTGACCGGGTTGGTTTTGAT

## Data Availability

All of the data supporting the reported results can be found in the manuscript and its Appendix A.

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
