# Peer review of "Effect of Thrombin on the Metabolism and Function of Murine Macrophages"

_cells, 2022, doi:10.3390/cells11101718_

Round 1

Reviewer 1 Report

Ukan and colleagues investigated the effect of thrombin on cultured macrophages using a wide variety of techniques. The polarization of macrophages and the influence of the coagulation pathway is of general interest to immunologist and vascular scientists. The manuscript is written well and the figures are very clear and expertly presented. However, I do have several minor comments before publication is possible.

Methods: Include the sequence of the housekeeping genes

Figure 1: Define HK, which I presume are housekeeping genes.

Section 3.4 and Figure 5: Please shortly introduce and explain the reasoning behind adding SMOC1 in this experiment. An M0 + SMOC1 control is needed in this experiment to show Thrombin-mediated effects.

In general:

Show the viability of the different conditions of macrophage polarization, possibly in supplementary materials.

Report the number of individual experiments that the data is representative of. Every experiment needs to have been repeated at least two times.

Genes should be noted using the official gene names and formatting, both in the figures and the methods. For example, CD206 is the protein and Mrc1 the gene.

Author Response

Reply to Reviewer #1

We would like to thank the reviewer for her/his constructive criticism of our manuscript. We hope that the additional data provided together with the changes made to the text satisfactorily address the concerns raised.

Methods: Include the sequence of the housekeeping genes

Reply: The details about the housekeeping genes have been added as requested to page 3, lines 106-109.

Figure 1: Define HK, which I presume are housekeeping genes.

Reply: Yes, the reviewer is 100% correct. The abbreviation has now been defined in the figure legend.

Section 3.4 and Figure 5: Please shortly introduce and explain the reasoning behind adding SMOC1 in this experiment. An M0 + SMOC1 control is needed in this experiment to show Thrombin-mediated effects.

Reply: We have highlighted our reasoning more clearly in the text and included new data that clearly show the impact of recombinant SMOC1 and a SMOC1 antibody on the phagocytic response induced by thrombin (see Figure 6 in the revised manuscript).

In general:

Show the viability of the different conditions of macrophage polarization, possibly in supplementary materials.

Reply: The reviewers request was rather a strange one. We can demonstrate viability by a number of methods including the phagocytosis assays shown in Figure 4 and 5. In order for us to conduct the analysis we assessed confluence at the end of the polarization period which also indicates viability. Furthermore, only viable macrophages are able to phagocytose pHrodo zymosan and oxLDL.

Report the number of individual experiments that the data is representative of. Every experiment needs to have been repeated at least two times.

Reply: Most of the experiments were performed at least twice and this information has been included in the figure legends as requested. The analysis of the metabolome and the RNA-seq data were all generated in on experiment with n=5-6 biologically independent samples. This is standard procedure for large screening experiments.

Genes should be noted using the official gene names and formatting, both in the figures and the methods. For example, CD206 is the protein and Mrc1 the gene.

Reply: We have checked the figures and text carefully to address this issue.

Reviewer 2 Report

Ukan and colleagues provide interesting data showing that thrombin alters macrophage metabolism and phenotype. Since thrombin is released during tissue injury, understanding the link between thrombin and innate immune response is of interest. The authors show that thrombin promotes the alternative activation of macrophages in culture and an increase in TCA metabolites. There are several concerns that are summarized below.

  1. The authors need to justify the dose of thrombin that is used in their study. Is this concentration physiological?
  2. The manuscript feels very disjointed. The authors touch on several key features of macrophages, i.e. metabolism, phenotype, phagocytosis. Only the surface is touched upon, there was little data to link the rational for determining these, and no mechanism is explored.
  3. It is unclear why the authors added the SMOC1 data in figure 5.
  4. It is unclear if macrophage phenotype alters metabolism or vise versa.
  5. Why was endothelial cell proliferation evaluated? Why is this important?
  6. In line 263, Figure 4D should be cited.
  7. The authors need to explain the loading control used for iNOS immunoblot.

Author Response

Reply to Reviewer #2

We would like to thank the reviewer for her/his constructive criticism of our manuscript. We hope that the additional data provided together with the changes made to the text satisfactorily address the concerns raised.

The authors need to justify the dose of thrombin that is used in their study. Is this concentration physiological?

Reply: The concentration of thrombin used in our study was 1 U/mL for up to 48 hours and is consistent with the fact that higher concentrations of thrombin can be detected in inflammatory settings.[1] This is also the highest concentration we used in our studies with platelets,[2] and was comparable with another study on macrophage function.[3] Other authors used much higher thrombin concentrations to address the impact of thrombin on monocyctes/macrophages i.e. 30 U/mL.[4] Also, studies focussing on monocytic cells with angiogenic potential (previously referred to as endothelial progenitor cells) used even higher concentrations i.e. 10 U/mL.[5] While most of the data generated in the revised manuscript were generated using 1 U/mL thrombin, the experiments shown in Figure 5 now show data generated using 0.1 and 1 U/mL.

The manuscript feels very disjointed. The authors touch on several key features of macrophages, i.e. metabolism, phenotype, phagocytosis. Only the surface is touched upon, there was little data to link the rational for determining these, and no mechanism is explored.

Reply: We agree completely with the reviewers comment and was attributable to the fact that some of the experiments that ran just before the deadline lacked our usual quality. The inclusion of much stronger data sets has helped us to pull it all together and as a consequence to rewrite the discussion.

It is unclear why the authors added the SMOC1 data in figure 5.

Reply: We appreciate the reviewer’s confusion which is related to the same issues as described above. SMOC1 is the first identified physiological activator of thrombin and increases thrombin-initiated events in platelets.2 Given the close association between monocytes/macrophages and platelets we determined whether or not this cell type could also be affected. The revised version of Figure 5 now shows clearly that SMOC1 also potentiates the thrombin response in macrophages while an antibody directed against thrombin attenuates it.

It is unclear if macrophage phenotype alters metabolism or vice versa.

Reply: The question of the chicken and the egg! The impact of metabolic reprogramming on macrophage phenotype has been explored extensively, and it seems safe to say that they affect each other.

Why was endothelial cell proliferation evaluated? Why is this important?

Reply: The rationale behind assessing endothelial proliferation was to evaluate the ability of thrombin-stimulated macrophages to promote angiogenesis. This is a classical action of alternatively activated or M2 macrophages which release high concentrations of angiogenic factors such as vascular endothelial growth factor as well as non-coding RNAs. The data obtained corroborate our hypothesis that thrombin polarizes macrophages to an alternatively activated phenotype that share some similarities with M2a-polarized cells and also possesses clear differences.

In line 263, Figure 4D should be cited.

Reply: We apologise for the mislabelling, which has been corrected.

The authors need to explain the loading control used for iNOS immunoblot

Reply: We used non muscle myosin (NMM) as a loading control for immunoblotting which is equally expressed in all macrophage phenotypes, and its expression is not affected through the process of polarization. The abbreviation has now been explained in the figure legend.

[1] Posma JJN, Posthuma JJ, Spronk HMH. Coagulation and non-coagulation effects of thrombin. J Thromb Haemost. 2016;14(10):1908-1916

[2] Delgado Lagos F, Elgheznawy A, Kyselova A, et al. Secreted modular calcium-binding protein 1 binds and activates thrombin to account for platelet hyperreactivity in diabetes. Blood. 2021;137(12):1641-1651

[3] White MJV, Gomer RH. Trypsin, tryptase, and thrombin polarize macrophages towards a pro-fibrotic M2a phenotype. PLoS One. 2015;10(9):e0138748.

[4] López-Zambrano M, Rodriguez-Montesinos J, Crespo-Avilan GE, Muñoz-Vega M, Preissner KT. Thrombin promotes macrophage polarization into M1-like phenotype to induce inflammatory Responses. Thromb Haemost. 2020;120(4):658-670.

[5] Tarzami ST, Wang G, Li W, Green L, Singh JP. Thrombin and PAR-1 stimulate differentiation of bone marrow-derived endothelial progenitor cells. J Thromb Haemost. 2006;4(3):656-663.

Reviewer 3 Report

The manuscript “Effect of thrombin on the metabolism and function of murine 2 macrophages” by Udan et al describes the novel role of thrombin on macrophage polarization leaning towards the distinct pro-resolving phenotype. The manuscript is potentially interesting and provides insights to the field of macrophage polarization and thrombin. Article is very well written and the conclusions drawn are supported by the results. I have few concerns before its publication.

  1. Authors should elaborate the introduction part which is very crisp. I would suggest to expand on the macrophage polarization and SMOC1 topics.
  2. What was the rationale behind choosing only MRC-1, MMP9 and CD3 as M2 markers? Why authors excluded other classical markers such as PPAR-?, CD163, HIF-1⍺, etc.,

Author Response

Reply to Reviewer #3

We would like to thank the reviewer for her/his constructive criticism of our manuscript. We hope that the additional data provided together with the changes made to the text satisfactorily address the concerns raised.

Authors should elaborate the introduction part which is very crisp. I would suggest to expand on the macrophage polarization and SMOC1 topics.

Reply: We appreciate the reviewer’s comment. We have kept the introduction brief but have expanded on the topic of SMOC1 in the discussion section of the revised manuscript.

What was the rationale behind choosing only MRC-1, MMP9 and CD3 as M2 markers? Why authors excluded other classical markers such as PPAR-?, CD163, HIF-1⍺, etc.

Reply: There are a plethora of markers that can be used to characterize macrophages and each group seems to have its favourites. We had the option of performing a larger RT-qPCR screen with a lot of general markers or looking at different aspects of macrophage polarization including metabolism and general changes in gene expression. Our original plan was to perform RNA-seq to assess the macrophage phenotype induced by thrombin in more detail but the results were not available before the submission deadline. RNA-seq data have now been included in the revised manuscript as Figures 5 and 6A, which renders the inclusion of additional RT-qPCR data unnecessary.